# Spatial variation of heart failure and air pollution in Warwickshire, UK: an investigation of small scale variation at the ward-level

Oscar Bennett,[1] Ngianga-Bakwin Kandala,[2,3] Chen Ji,[2] John Linnane,[1] Aileen Clarke[2]

[1]Public Health Warwickshire, Communities Group, Warwickshire County Council, Warwick, UK
[2]Division of Health Sciences, University of Warwick Medical School, Coventry, UK
[3]Division of Epidemiology and Biostatistics, School of Public Health, University of Witwatersrand, Johannesburg, South Africa

**Correspondence to**
Dr Ngianga-Bakwin Kandala;
N-B.Kandala@warwick.ac.uk

## ABSTRACT

**Objectives:** To map using geospatial modelling techniques the morbidity and mortality caused by heart failure within Warwickshire to characterise and quantify any influence of air pollution on these risks.

**Design:** Cross-sectional.

**Setting:** Warwickshire, UK.

**Participants:** Data from all of the 105 current Warwickshire County wards were collected on hospital admissions and deaths due to heart failure.

**Results:** In multivariate analyses, the presence of higher mono-nitrogen oxide (NOx) in a ward (3.35:1.89, 4.99), benzene (Ben) (31.9:8.36, 55.85) and index of multiple deprivation (IMD; 0.02: 0.01, 0.03), were consistently associated with a higher risk of heart failure morbidity. Particulate matter (Pm; −12.93: −20.41, −6.54) was negatively associated with the risk of heart failure morbidity. No association was found between sulfur dioxide ($SO_2$) and heart failure morbidity. The risk of heart failure mortality was higher in wards with a higher NOx (4.30: 1.68, 7.37) and wards with more inhabitants 50+ years old (1.60: 0.47, 2.92). Pm was negatively associated (−14.69: −23.46, −6.50) with heart failure mortality. $SO_2$, Ben and IMD scores were not associated with heart failure mortality. There was a prominent variation in heart failure morbidity and mortality risk across wards, the highest risk being in the regions around Nuneaton and Bedworth.

**Conclusions:** This study showed distinct spatial patterns in heart failure morbidity and mortality, suggesting the potential role of environmental factors beyond individual-level risk factors. Air pollution levels should therefore be taken into account when considering the wider determinants of public health and the impact that changes in air pollution might have on the health of a population.

## Strengths and limitations of this study

- The model employed a fully Bayesian approach using Markov Chain Monte Carlo (MCMC) techniques for inference and model checking.[11,12]
- A single air pollution measurement in 2010 was used and it was assumed that there was no significant change in this value over the 2005–2013 periods that mortality and hospital admission data were gathered from.
- The cross-sectional nature of the study, unfortunately, does not allow us to establish temporality and thus clearly demonstrate causality of the observed associations.
- This was an ecological study, analysing characteristics and risk at the ward rather than individual level. Consequently, associations and conclusions found from the aggregate data may not be directly applicable to individuals.

## INTRODUCTION

Air pollution has been linked to the development and exacerbation of a number of health problems. This link seems especially clear for cardiovascular and respiratory diseases such as ischaemic heart disease, heart failure, asthma, influenza and lung cancer.[1–4] It is well established that the ambient levels of air pollution in a region can have an impact on the health status of the population which inhabits it.[5] Air pollution levels should therefore be taken into account when considering the wider determinants of public health and the impact that changes in air pollution might have on the health of a population.

Warwickshire is an English county within the UK approximately 112 km North-West of London. The issue of air pollution has been highlighted in Warwickshire recently by the setting up of Air Quality Management Areas (AQMAs) in a number of different parts of the county. These are specific areas in the county that have been identified as places where, without a focused local council strategy to reduce air pollution, future government

targets for air pollutant concentrations may not be met. Within Warwickshire, the specific air pollutant that has been identified as a problem and that led to these special areas being setup is mono-nitrogen oxide (NOx).[6]

The objective of the present study was, therefore, to look for any relationship between levels of air pollution and the morbidity and mortality associated with heart failure within Warwickshire in the past few years (2005–2013 for morbidity, 2007–2012 for mortality). We examined a range of traditional air pollutants for an ecological association with heart failure morbidity and mortality at the county level. Furthermore, the present analysis attempted to highlight spatial patterns in heart failure morbidity and mortality risk within the county, after multiple adjustments for proximate, county-level factors. The geographic locations of wards within the county can be considered as proxy measures of many other unmeasured factors such as availability and access to health services, individual health-seeking behaviour, preventive ward policy and general ward factors. Such estimates might illustrate how much can be learned by detailed exploratory analyses as well as how these data can be used to strategically inform policy aiming at the prevention and management of serious conditions such as heart failure in these settings.

## THE EVIDENCE FROM PUBLISHED LITERATURE FOR THE LINK BETWEEN AIR POLLUTION AND HEALTH PROBLEMS

Studies have shown an effect on the health of populations caused by long-term exposure as well as short-term 'spikes' in local air pollution levels.[5]

A relevant systematic review and meta-analysis has been published recently in *The Lancet*.[2] It pooled the results of studies that have been conducted worldwide looking at the temporal relationship between the levels of a number of different air pollutants with local heart failure hospital admission rates and mortality rates. This showed a clear link between short-term rises in all types of air pollution (except ozone) and rises in hospital admissions and mortality due to heart failure. This study did not look at any potential effects from long-term exposure to air pollution. However, it did provide strong evidence that air pollution can 'exacerbate' heart failure, increasing the likelihood that a patient with existing heart failure will become sufficiently unwell to require hospital admission, or even die.

The ESCAPE study[3] published recently in the *Lancet Oncology*, pooling the results of 17 cohort studies from around Europe, looked at the ambient levels of air pollution in an area (Pm and NOx) and the incidence of lung cancers in the inhabitants of that area during many years of follow-up (mean 12.8 years). This was intended to look at the risk associated with long-term exposure to air pollution. A statistically significant correlation was found between the levels of particulate matter air pollution and the local incidence of lung cancer diagnoses. Pm with a diameter of less than 10 μm had a HR of 1.22

(95% CI 1.03 to 1.45) per (10 μg/m$^3$). In other words, for every increase in particulate matter pollution of 10 μg/m$^3$ there was a corresponding immediate increase in the chance of being diagnosed with lung cancer of 22% (95% CI 3% to 45%). There was also a correlation between traffic volume within 100 m of a residence and a modest increase in the rate of lung cancer (HR 1.09 (CI 0.99 to 1.21)). No similar correlation was found with NOx air pollution. This suggests that long-term exposure to higher levels of particulate matter air pollution may increase the incidence of lung cancer in a population.

Another study published in the USA in 2004 looked at the long-term effect of exposure to particulate matter air pollution and the mortality attributed to different cardiovascular and respiratory diseases in different areas of the USA.[4] A good correlation was found between the degree of long-term exposure to Pm air pollution and increases in mortality from cardiovascular diseases, including heart failure. Interestingly, this was not found to be the case for most respiratory diseases. There was also an element of the study that looked at the effect of a person's smoking status on the mortality statistics. This found, as expected, a strong link between mortality and smoking. However, it also found that air pollution contributed additional cardiovascular mortality risk on top of that attributed to smoking. This was at least an additive, if not a synergistic effect.

## METHODS
### Study data

In order to carry out this project, data were collected on the geographical distribution of air pollution within Warwickshire. These data included information about each of four individual components of air pollution (NOx, sulfur dioxide, particulate matter and benzene), which could then be united into a combined index (all of the contributions added together). A single recorded level from 2010 of each air pollutant for each ward was used in the study. This was then compared to collected data about:

1. The geographical distribution of home addresses of patients who were admitted to hospital because of heart failure or a complication of heart failure. Hospital admission rate in an area due to heart failure was used as a proxy indicator for the level of heart failure morbidity within that area.
2. The geographical distribution of home addresses of patients who died from heart failure, or whose death was contributed to by heart failure.

These data were collected by the Warwickshire Observatory, which is part of the Warwickshire County Council in charge of collecting and handling statistics relating to the county. Mortality data for the analysis were supplied via the Warwickshire Public Health Intelligence Team and was sourced from the Public Health Mortality Files, Office for National Statistics.

Hospital admissions data were accessed via the Ventris Business Intelligence System, Arden Commissioning Support Unit.

Ward level population data were obtained from the 2011 census. Warwickshire is divided into 105 wards. Data from all of the 105 current Warwickshire County wards were collected.

The potential confounding variables of, first, age structure of a population within a ward and, second, levels of social deprivation within a ward were identified.[7] [8] These were then controlled for in the second stage of the statistical analysis (see below).

Information about the age structure of wards was obtained from the Office of National Statistics' mid-2010 estimates (obtained from the Warwickshire Observatory website).[9] The representative statistical value of 'percentage (%) of population above age 50' was used as an indicator of wards with a higher proportion of older people.

Information about social deprivation was obtained from the English Indices of Deprivation published by the Department for Communities and Local Government.[10] The index of multiple deprivation (IMD) averaged across each ward was used as an indicator of the level of deprivation within the wards of Warwickshire.

The information on crude observed air pollution level distribution, heart failure hospital admission rates and mortality rates were then represented on maps of Warwickshire (figure 1).

## Statistical analysis

To account for spatial autocorrelation in observed heart failure hospital admission and mortality rates at the ward level in Warwickshire we applied a unified approach to account for possible air pollution effects of environmental risk factors. This was achieved using a geoadditive semiparametric mixed model. The model employed a fully Bayesian approach using Markov Chain Monte Carlo (MCMC) techniques for inference and model checking.[11] [12] Response variables were defined as the count (per 1000 population) of heart failure morbidity or mortality in a ward (Poisson model): $y_i | \eta_i, \delta \sim B(\eta_i, \delta)$ for a binomial formulation.

(Poisson model):

$$y_i | \eta_i, \delta B(\eta_i, \delta) \qquad (1)$$

for a binomial formulation and a geoadditive semiparametric predictor $\mu_i = h(\eta_i)$:

$$\eta_i = f_1(x_{i1}) + \cdots \cdots f_p(x_{ip}) + f_{spat}(S_i) + \varepsilon_i \qquad (2)$$

where h is a known response function with a poison link function, $f_1, \ldots, f_p$ are non-linear smoothed effects of the metrical covariates (time in years), and $f_{spat}(s_i)$ is the effect of the spatial covariate $s_i \in \{1, \ldots, S\}$ labelling the ward in Warwickshire. Regression models with predictors such as those in equation 2 are sometimes referred to as

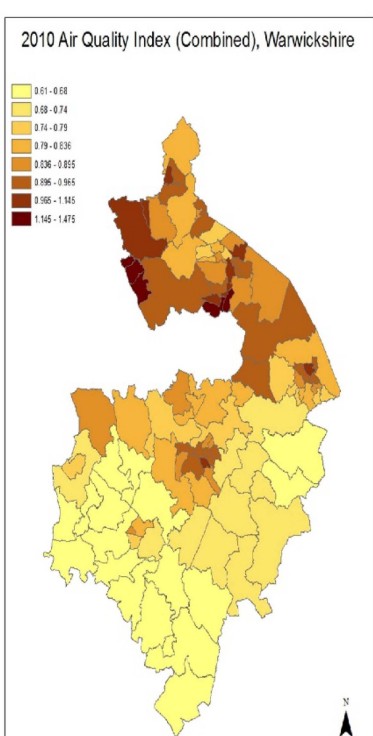
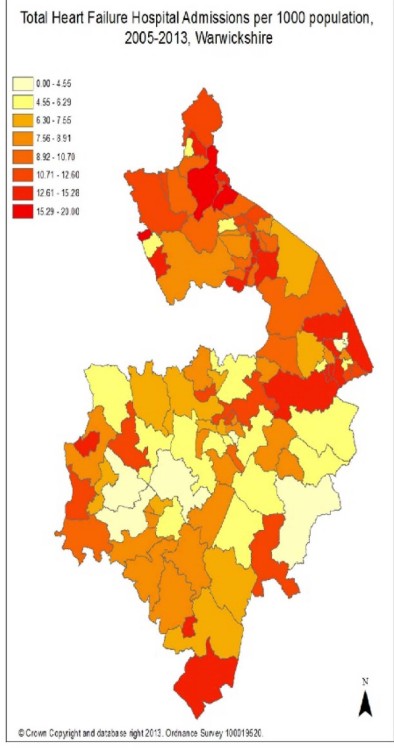
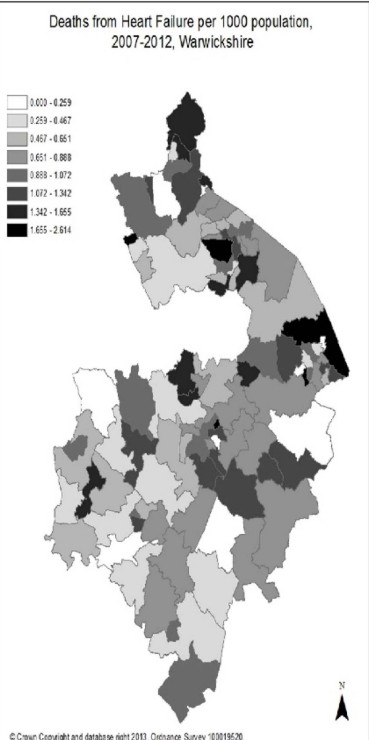

**Figure 1** Warwickshire map with 2010 air quality index (all components of air pollution combined) displayed by ward (left) Warwickshire map with number of heart failure hospital admissions per 1000 population between 2005 and 2013 displayed by ward (centre) and Warwickshire map with total heart failure deaths per 1000 population between 2007 and 2012 displayed by ward (right).

geoadditive models. P-spline priors were assigned to the functions f1,…,f$_p$, non-informative priors were used for fixed effects parameters and a Markov random field prior was used for f$_{spat}$ (s$_i$). More detailed information about the modelling approach can be found elsewhere.[11 13] The standard measure of effect was the posterior mean (PM) and 95% credible region (CR).

The analysis was carried out using V.2.0.1 of the BayesX software package, which permits Bayesian inference based on MCMC simulation techniques.[13] Multivariate Bayesian geoadditive regression models were used to evaluate the significance of the PM determined for the fixed effects and spatial effects between air pollution and the morbidity and mortality from heart failure within Warwickshire. Each component of air pollution was looked at separately and then the combined index in unadjusted models was examined. Next, fully adjusted multivariate Bayesian geoadditive regressions analyses were performed to look again for statistically significant correlations between these variables, but this time further controlling for any influence from age structure or social deprivation.

## RESULTS

Figures 1 and 2 display the observed data collected for this study on maps using graduated colouring to represent data value categories within each ward.

### Observed air quality map

Figure 1 (left) was produced using 2010 data from the Warwickshire Observatory. It is based on a combined air quality indicator, which is a combination of information about the contribution to air pollution from NOx, sulfur dioxide, particulate matter and benzene. The Warwickshire Observatory description of this index reads as follows:

> Combined Air Quality Indicator (estimates of emissions for four pollutants: benzene, nitrogen dioxide, sulfur dioxide and particulates) for small areas (modelled to 1 km grid squares) where an index value of 1 is equivalent to the national standard for each pollutant. The values are then summed so an overall score of 4 would represent all four pollutants being present at the national standard level.[14] These specific standards are described in the Air Quality Standards Regulations 2010.

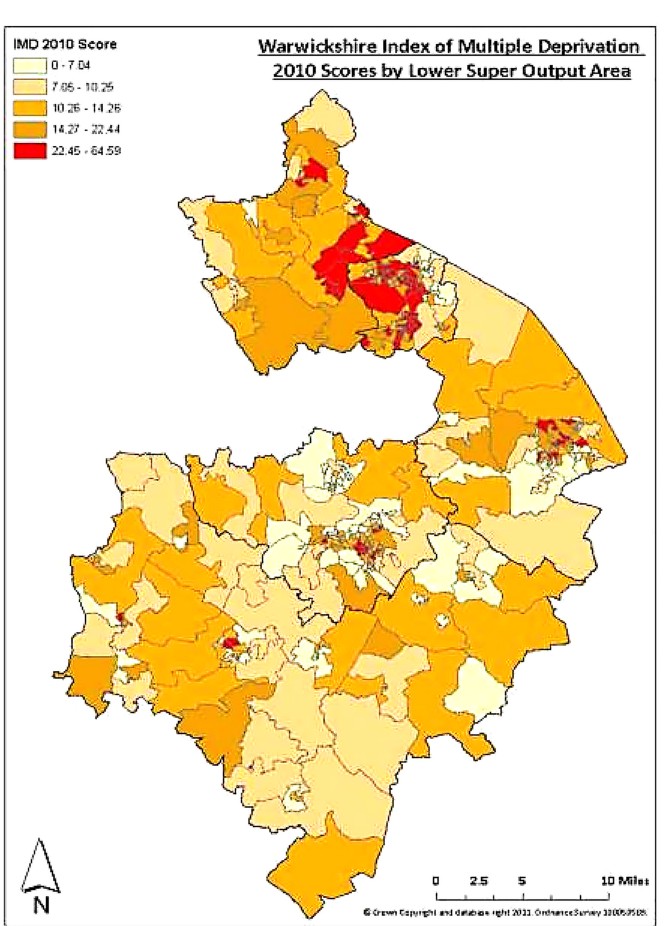
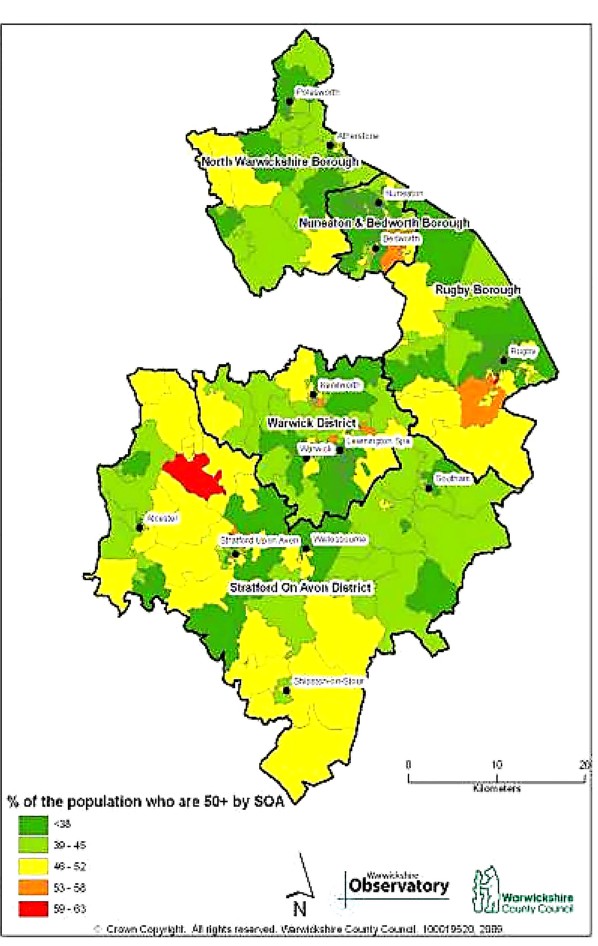

**Figure 2** Warwickshire map displaying the 2010 levels of deprivation (expressed as Multiple Deprivation Scores) by LSOAs. Produced by the Warwickshire Observatory (left) and Warwickshire map displaying the 2008 mid-year estimates of percentage of people over the age of 50 by SOA. Produced by the Warwickshire Observatory (right).

The geographical pattern of the observed air pollution across the county shows a higher level of air pollution near the more urban centres of Birmingham and Coventry. The proximity of parts of the county to motorways such as the M6 and M42 could also be a contributing factor to the observed pattern. There was no place in the county where the level of any pollutant exceeded the national standard in 2010.

### Heart failure hospital admission map

Figure 1 (centre) shows the geographical distribution of the density of home addresses of patients admitted to hospital with a diagnosis of heart failure (or exacerbation of heart failure) within the April 2005–April 2013 period.

### Heart failure mortality map

Figure 1 (right) shows the geographical distribution of the density of home addresses of patients who died either directly or in part from heart failure in the 2007–2012 (inclusive) periods.

Table 1 (left panel) displays posterior means of heart failure admission across the selected covariates following multivariate Bayesian geoadditive regression analyses.

The PM and 95% CR of overall hospital admission rates due to heart failure and mortality rates from heart failure were (6.19 (3.84, 8.97)) and (4.13 (1.18, 7.43)) (table 1), respectively. On average, the presence of a higher NOx indicator in a ward (PM and 95% CR: 3.35 (1.89, 4.99)), benzene indicator (Ben; 31.9 (8.36, 55.85)) and IMD 2010 score (0.02 (0.01, 0.03)), were consistently associated with higher risk of heart failure morbidity. The particulates indicator (Pm; −12.93 (−20.41, −6.54)) was negatively associated with the risk of heart failure morbidity. Sulfur dioxide indicator ($SO_2$) was not associated with heart failure morbidity.

Table 1 (right panel) shows the same corresponding results for heart failure mortality. This table shows that the risk of heart failure mortality was higher in wards with a higher NOx (4.30 (1.68, 7.37)) and wards with more inhabitants over 50 years old (1.60 (0.47, 2.92)).

The particulates indicator (Pm) was negatively associated with heart failure mortality. The sulfur dioxide indicator ($SO_2$), benzene indicator (Ben) and IMD score were not associated with heart failure mortality.

The combined air pollution index (all indicators averaged together) when incorporated into a separate model combining age and social deprivation was significantly positively associated with heart failure morbidity (1.39 (0.87, 1.81)) as well as mortality (1.79 (0.85, 2.55)) across the county.

In figures 3 and 4, the left-hand maps show the unadjusted estimates of posterior total residual ward means of heart failure morbidity and mortality, respectively. In figures 5 and 6, the left-hand maps show the adjusted PM after multiple adjustment for the geographical location, taking into account the autocorrelation structure in the data, the uncertainty in the ward level and all ward-level risk factors (air pollution components, age, social deprivation) for heart failure morbidity and mortality, respectively. The red colour indicates the maximum posterior mean recorded while green denotes the lowest mean. The right-hand maps in figures 3–6 show the 95% posterior probability of heart failure and mortality, which indicate the statistical significance associated with the total excess risk. Black colour indicates a negative spatial effect (associated with increased risk of heart failure and mortality), white colour a positive effect (a decreased risk) and grey colour a non-significant effect.

In general, there was consistently higher heart failure morbidity risk in northern wards, particularly around Nuneaton and Bedworth, and lower heart failure morbidity risks in the southern wards, particularly within the district of Stratford-on-Avon. However, all this variation could partially be accounted for within the model generated within this study (taking into account air pollution, age and social deprivation levels).

Heart failure mortality risk was, in general, higher in northern wards around Nuneaton and Bedworth as well as in some central areas around Warwick, Royal Leamington Spa and Kenilworth. Heart failure mortality risk was again lower in more southern wards particularly within

**Table 1** Posterior mean (PM) of fixed effects estimates of heart failure admission and mortality across air pollution indicators (Warwickshire 2005–2013)

| Variable | Heart failure admission PM and 95% CI* | Heart failure mortality PM and 95% CI* |
|---|---|---|
| Constant | 6.19 (3.84 to 8.97) | 4.13 (1.18 to 7.43) |
| Nitrogen dioxide indicator ($NO_2$) | 3.35 (1.89 to 4.99) | 4.30 (1.68 to 7.37) |
| Sulfur dioxide indicator ($SO_2$)) | 7.75 (−3.84 to 17.99) | 11.02 (−1.21 to 22.41) |
| Particulates indicator (Pm) | −12.93 (−20.41 to −6.54) | −14.69 (−23.46 to −6.50) |
| Benzene indicator (Ben) | (31.9:8.36 to 55.85) | 25.98 (−6.62 to 54.22) |
| Over 50 years % | 0.70 (−0.63 to 1.98) | 1.60 (0.47 to 2.92) |
| IMD 2010 score | 0.02 (0.01 to 0.03) | 0.00 (−0.01 to 0.01) |

*Spatially adjusted posterior mean (PM) from Bayesian geoadditive regression models after controlling for fixed effect of all air pollutions indicators: mono-nitrogen oxide (NOx), sulfur dioxide ($SO_2$), particulate matter (Pm), benzene (Ben) and combined index and the county of residence (spatial effects).
IMD, index of multiple deprivation.

**Figure 3** Left: unadjusted total residual spatial effects of morbidity risk associated with heart failure at ward level in Warwickshire; shown are the posterior means. Right: corresponding posterior probabilities at 80% nominal level. Red coloured—high risk; Green coloured—low risk; Black coloured—significant positive spatial effect; White coloured—significant negative spatial effect; Grey coloured—no significant effect.

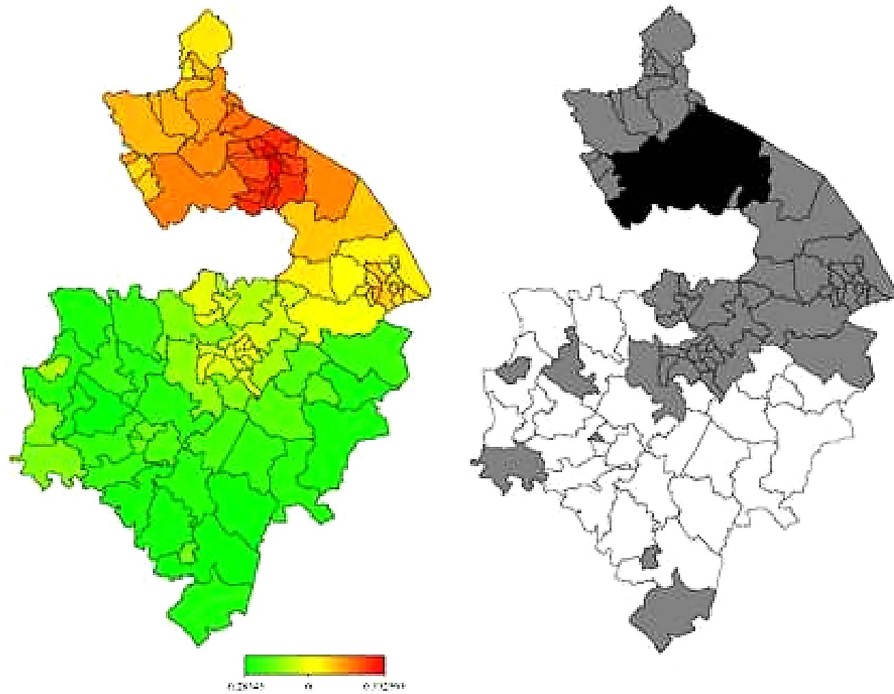

Stratford-on-Avon. Unlike with morbidity, however, our model could not explain all this variation in heart failure mortality. Even after taking into account air pollution, age and social deprivation, the rate of death from heart failure remained significantly higher than would be expected in areas within and around Nuneaton, Bedworth, Warwick, Royal Leamington Spa and Kenilworth.

In sensitivity analyses, we tested several models and our results were not substantially altered after removing one or two pollutants (data not shown).

## DISCUSSION

Before even considering air pollution, this study helps to demonstrate the inequality of risk from heart failure disease and death that exists for individuals living in different parts of the county of Warwickshire. There is a significant excess risk of disease as well as death in more northern wards within and surrounding Nuneaton and Bedworth. Much (but not all) of this variation could be attributed to the air pollution, age structure and social deprivation that exists in these areas according to our

**Figure 4** Left: unadjusted total residual spatial effects of mortality risk associated with heart failure at ward level in Warwickshire; shown are the posterior means. Right: corresponding posterior probabilities at 80% nominal level. Red coloured—high risk; Green coloured—low risk; Grey coloured—no significant effect; Black coloured—significant positive spatial effect; White coloured—significant negative spatial effect.

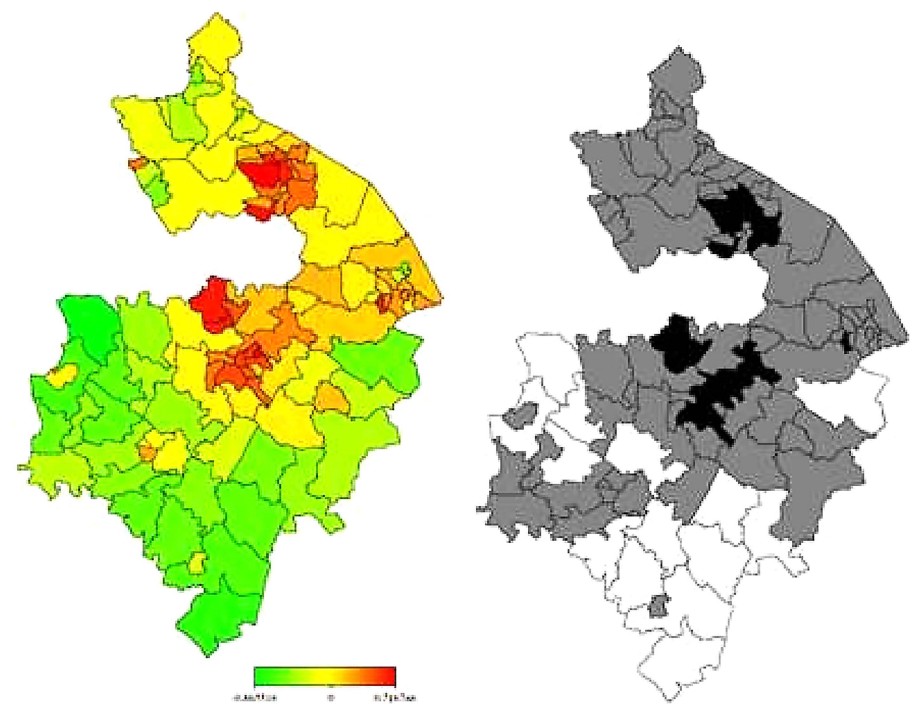

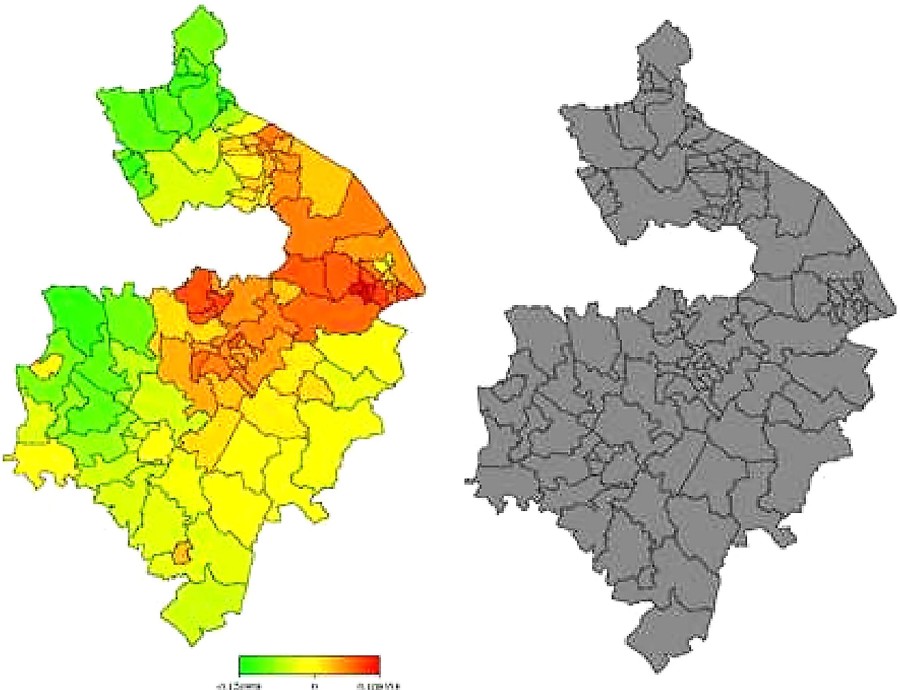

**Figure 5** Left: total residual spatial effects of morbidity risk associated with heart failure at ward level in Warwickshire; shown are the posterior means of the full model (IMD 2010, Over 50 and the 4 indicators of air pollution). Right: corresponding posterior probabilities at 80% nominal level. Red coloured—high risk; Green coloured—low risk; Black coloured—significant positive spatial effect; White coloured—significant negative spatial effect; Grey coloured—no significant effect.

model. Measures that seek to address air pollution and social deprivation could be expected therefore to help mitigate against cardiovascular risks within these local populations.

The present study corroborates the notion that air pollution is an increasingly important public health issue in Warwickshire. Higher levels of the average air pollution index looked at in this study correlated significantly with increased levels of heart failure morbidity as well as mortality across the county, even after removing the effects of age structure and social deprivation. The individual component of NOx air pollution, particularly, seems to

contribute risk to the morbidity and mortality of heart failure within the county, suggesting that it may have a particularly detrimental effect on heart failure patients. This reinforces the importance of the AQMAs set up within the county in response to high-NOx levels. Road traffic is a large contributor to air pollution within Warwickshire. Diesel engines are responsible for a large part of the NOx component of these emissions.

An unexpected result also appeared within our analysis. Particulate matter air pollution became significantly negatively correlated with heart failure morbidity and also mortality when incorporated into our model, with

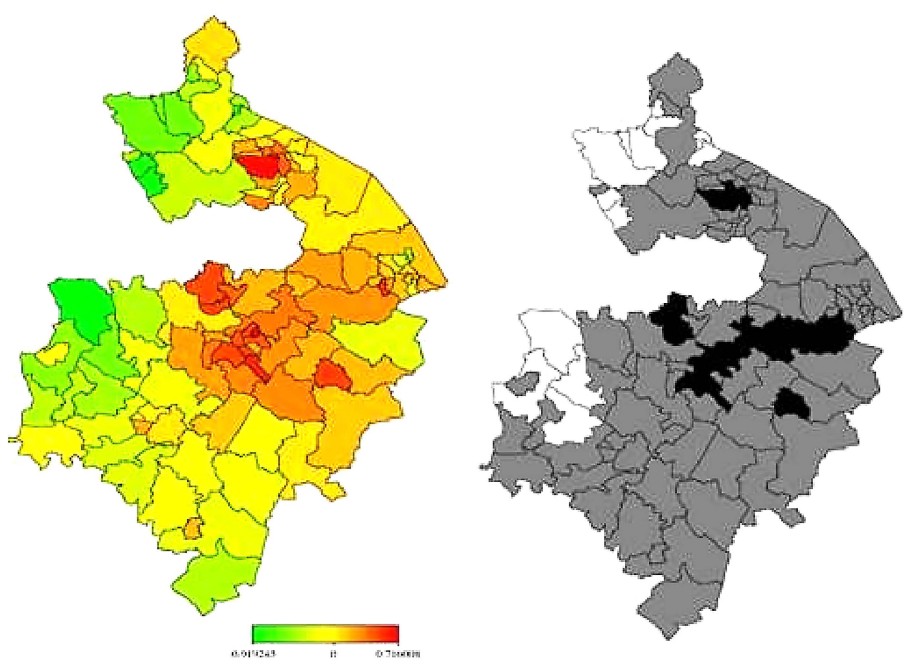

**Figure 6** Left: total residual spatial effects of mortality risk associated with heart failure at ward level in Warwickshire; shown are the posterior means of the full model (IMD 2010, Over 50 and the 4 indicators of air pollution). Right: corresponding posterior probabilities at 80% nominal level. Red coloured—high risk; Green coloured—low risk; Black coloured—significant positive spatial effect; White coloured—significant negative spatial effect; Grey coloured—no significant effect.

all risk factors taken into account and controlled for in this study. This would imply some sort of unexpected 'protective influence' from Pm air pollution on heart failure patients. This clearly contradicts our expectations and is at odds with a wealth of existing evidence that indicates that Pm air pollution contributes risk to and exacerbates cardiovascular disease.[4]

We offer a possible explanation for this based on the following four observations:

1. The aforementioned negative correlation of Pm air pollution with heart failure morbidity and mortality in our model.
2. Pm air pollution actually varied very little across the county compared to the other types of air pollution. All types of air pollution tended to decrease in rural areas, but Pm tended to decrease much *less* compared to the other components of air pollution. Consequently, in rural areas of the county where most types of air pollution are significantly lower, Pm pollution was relatively higher compared to NOx, Benzene and $SO_2$.
3. There seems to be a high risk of heart failure deaths in urban centres (particularly Nuneaton, Bedworth, Warwick, Royal Leamington Spa and Kenilworth), higher than can be explained by our model.
4. Conversely, there seems to be a particularly low risk of heart failure deaths in some rural areas within the western part of Stratford-on-Avon, lower than can be explained by our model.

A possible hypothesis based on these observations is that there is an additional factor influencing the morbidity and mortality of heart failure not looked at in this study, namely the urban/rural nature of a patient's living environment. It could be the case that living in an urban environment contributed risk and living in a rural area provided protection against heart failure morbidity and mortality. This would be an effect in *addition* to any increase in air pollution or social deprivation within urban settings compared to rural settings. This could certainly be plausible in principle, with people in rural areas perhaps doing more physical activity, eating more healthily, etc. If this were the case it would explain the excess deaths in urban centres found in this study. It could also be responsible for the unexpected protective factor attributed to Pm air pollution in our analysis. Given that the Pm component of air pollution is *relatively* higher than the other components in rural areas, the protection that living in rural areas affords individuals could be misleadingly attributed (in our statistical analysis) to the particulate matter component of air pollution present in those areas. Further work will need to be carried out looking into this possible link between urban/rural living environments and heart failure morbidity and mortality. It could be very revealing to carefully characterise this effect if it indeed exists, as it may be an indication of unrecognised cardiovascular risk/protective factors associated with urban/rural living that exist within Warwickshire.

However, it is also important to bear in mind that this is an ecological study and all the relationships picked up between variables in this study have been found using aggregate data at the ward level (number over 50 years of age, average IMD score, average air pollution across ward, overall numbers of deaths and hospital admissions due to heart failure). It is not always a trivial task to extrapolate the conclusions drawn from such a study down to the level of individuals. Such a task would involve drilling down to individual level data and repeating the analysis, a task that was beyond the scope of this particular study. It is possible that the unexpected negative correlation between particulate matter air pollution and heart failure could disappear when data are analysed at the individual level—an example of an ecological fallacy. Consequently, it would be prudent to regard the results from the individual components of air pollution with cautious interest rather than viewing them as proof of any real effects.

However, despite these caveats, this study has been able to provide some helpful information at the population level worthy of consideration. A health inequality has been revealed, and the manner in which this inequality is influenced by age, social deprivation and the combined index of air pollution has been demonstrated. Such information should help inform policy decisions that would influence society at a population level and hopefully improve public health in the long run.

There are some limitations in this study worth considering that result from assumptions made along the way. A single air pollution measurement in 2010 was used and it was assumed that there was no significant change in this value over the 2005–2013 periods that mortality and hospital admission data were gathered from. The resulting cross-sectional nature of the study does not allow establishing temporality and thus causality of the observed associations. There was also no way to determine the length of time that individual members of the population within a ward had lived in that area, and thus how long they had been exposed to the measured ambient air pollution level. It was assumed that people with home addresses in a ward were exposed significantly to the levels of air pollution in that ward. Finally, as already mentioned, this was an ecological study, using aggregate data of risk factors to look for associations with aggregate data of morbidity and mortality. It is not always possible to apply such associations from a population level down to the individuals within that population.

In summary, this study has provided a number of interesting results. First, it has helped to quantify and map the inequality that exists across different parts of Warwickshire with regard to heart failure risk. It has also provided some interesting circumstantial evidence of a link between heart failure morbidity and air pollution. Finally, it has also given a suggestion of a possible link between living in urban environments and a higher risk of cardiovascular disease and a corresponding lower risk

from living in rural environments. More work will need to be carried out to look into this particular possibility. It would be informative to run this type of analysis while factoring in the influence of a person's distance from their nearest urban centre. This urban/rural factor should be further explored and mined for additional information as it could be an indication of hitherto unconsidered factors influencing the health status of the population of Warwickshire and possibly further afield.

In order to determine the validity of our conclusions at the individual level, further work would need to be carried out analysing the available data from individual patients (risks and outcomes). Such work could help to characterise the true effect of different components of air pollution at the individual level. It would also be interesting to determine if the different components of air pollution act as effect modifiers on each other.

It would be possible to look at the effects of air pollution variation in the shorter term as well. For example, looking at how local 'spikes' in air pollution affect the rates of hospital admissions locally immediately following it. This could be carried out in Leamington Spa where there is an air quality monitoring station constantly measuring the levels of air pollutants.

Other health problems, such as ischaemic heart disease and respiratory diseases, have been linked with air pollution as well and it could be informative to also look into these links locally.

**Contributors** OB conceived the idea, analysed the data, contributed to formulating the results and wrote the first draft. N-B K analysed the data, advised on statistical aspects, contributed to formulating the results and wrote the second draft. CJ analysed the data. JL helped coordinate the project and cowrote the final draft. AC coordinated the project, advised on all aspects and cowrote the final draft.

**Funding** This paper presents independent research supported by the National Institute for Health Research (NIHR) Collaborations for Leadership in Applied Health Research and Care West Midlands. The views expressed are those of the authors and not necessarily those of the NHS, the NIHR or the Department of Health.

**Competing interests** None.

**Provenance and peer review** Not commissioned; externally peer reviewed.

**Data sharing statement** No additional data are available.

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
