## [Reviewer comments · BMJ Open]

ARTICLE DETAILS

TITLE (PROVISIONAL)	Spatial variation of Heart Failure and Air Pollution in Warwickshire, UK: An investigation of small scale variation at the ward-level
AUTHORS	Bennett, Oscar; Kandala, Ngianga-Bakwin; Ji, Chen; Linnane, John; Clarke, Aileen

VERSION 1 - REVIEW

REVIEWER	Anoop Shah Farr Institute of Health Informatics Research University College London
REVIEW RETURNED	22-Jul-2014

GENERAL COMMENTS	Data were analysed using a Bayesian approach, so the report should state what priors were used and why. The key limitation is that this is an ecological study and data from individuals within wards is required to confirm these findings. This may be possible through linkage of patients' general practice records with geographic data on pollution. The association between lower particulate pollution and higher incidence of heart failure may be spurious and may not be present in an individual patient analysis. The wording of some of the findings in the results could be improved, e.g. 'Particulate matter was positively non-significantly correlated with ...' is confusing; if the association is not statistically significant and the study has sufficient power to detect an important difference, use wording such as 'there was no significant evidence ...' instead. References to figures in the main text should follow conventional style for articles, e.g. 'Figure 1 shows ...' rather than 'This map'. Table 1 - What is the interpretation of the figures in the table. Are they rates? Figure 1 caption - missing closing bracket) after left Figure 2 - please provide more detail in the legend for figure 2 - what strength of effect (quantitatively) do the colours represent? Figures 3-5 - Grammar needs to be corrected in a few places, e.g. 'Shown is the posterior means ...'.
--

REVIEWER	Thomas N O Achia Division of Epidemiology and Biostatistics, School of Public Health, University of Witwatersrand, Johannesburg-South Africa
-----------------	--

GENERAL COMMENTS

The paper presents an interesting discussion concerning the association between air pollution and cardiovascular disease. The authors use a Bayesian Hierarchical spatial modeling approach to analyse the data providing an interesting way of looking at the problem.

There are a few issues that need comment.

In the abstract section under primary and secondary measures, it is not clear to me what the primary outcome(s) or predictor(s) were from the statement "Air pollution, heart failure hospital admissions and mortality data within Warwickshire were" mean.

The statement "Pm [-12.93:-20.41, -6.54] was negatively associated with the risk of heart failure morbidity but with no association with So2" in the abstract is rather confusing. Are the authors suggesting that Pm was associated with So2? Currently this is what the sentence suggests and needs revision.

In the abstract: "There was a striking variation in heart failure morbidity and mortality risk across wards, the highest risk being in the regions around Nuneaton and Bedworth." Please clarify what the word "striking" means

The comment: "Particulate matter (Pm) was negatively associated with the risk of heart failure morbidity" needs unpacking. Does this mean that presence of Pm is protective. If so, why?

The authors have provided a discussion around this but I am not very convinced. Could one or more of the other predictors be acting as an effect modifier? Did the authors investigate this by fitting two-way or other interaction terms? The attempt to justify the finding is not satisfactory.

There are issues here of "ecological fallacy" in nature that need to be reexamined.

Air pollution contributed an "additional rise" in the sentence just before the Methods section, does not make for good grammar.

The first sentence under the study data section should read: "In order to carry out this project, data was collected about the geographical distribution of air pollution within Warwickshire" as the placement of the comma after the word project gives a completely different meaning.

It could be helpful for the authors to clarify what each of the variables or symbols in the Binomial distribution presented on line 1-11 of page 8 are.

VERSION 1 – AUTHOR RESPONSE

Reviewer Name: Anoop Shah

Institution and Country Farr Institute of Health Informatics Research

University College London

Please state any competing interests or state 'None declared': None declared

Data were analysed using a Bayesian approach, so the report should state what priors were used and why.

Reply: The reviewer is right. We did not insert details information about the Bayesian modelling approach as we thought that it was behind the scope of this paper. However, have inserted the information requested by the reviewer and refer the readers to references where they can find more information about the modelling approach. (See page last paragraph of page 8 to page 9. Also see below :)

'(Poisson model): $y_i | \eta_i, \delta \sim B(\eta_i, \delta)$ (1) for a binomial formulation and a geo-additive semi-parametric predictor $\mu_i = h(\eta_i)$:

$$\eta_i = f_1(x_{i1}) + \dots + f_p(x_{ip}) + f_{\text{spat}}(s_i) + \xi_i \quad (2)$$

where h is a known response function with a poisson link function, f_1, \dots, f_p are non-linear smoothed effects of the metrical covariates (time in years), and $f_{\text{spat}}(s_i)$ is the effect of the spatial covariate $s_i \in \{1, \dots, S\}$ labelling the ward in Warwickshire. Regression models with predictors such as those in equation 2 are sometimes referred to as geo-additive models. P-spline priors were assigned to the functions f_1, \dots, f_p , and a Markov random field prior was used for $f_{\text{spat}}(s_i)$. More detailed information about the modelling approach can be found elsewhere (11, 13).

The key limitation is that this is an ecological study and data from individuals within wards is required to confirm these findings. This may be possible through linkage of patients' general practice records with geographic data on pollution. The association between lower particulate pollution and higher incidence of heart failure may be spurious and may not be present in an individual patient analysis.

Reply: We appreciate the point made by the reviewer and we agree with the reviewer in saying that the association between lower particulate pollution and higher incidence of heart failure may be spurious and may not be present in an individual patient analysis. However, the assertion of the reviewer is only speculative as we do not have access to the individual-level data, in which we can investigate the above hypothesis. We will endeavour to get access to the individual level data to test the above hypothesis. Thus, our far as our study is concerned, we have made it clear that the association found is an ecological association, which is bound to several limitations: causality cannot be inferred.

The wording of some of the findings in the results could be improved, e.g. 'Particulate matter was positively non- significantly correlated with ...' is confusing; if the association is not statistically significant and the study has sufficient power to detect an important difference, use wording such as 'there was no significant evidence ...' instead.

Reply: The wording in some places has been amended to improve clarity

References to figures in the main text should follow conventional style for articles, e.g. 'Figure 1 shows ...' rather than 'This map'.

Reply: This has been fixed.

Table 1 - What is the interpretation of the figures in the table. Are they rates?

Figure 1 caption - missing closing bracket) after left Figure 2 - please provide more detail in the legend for figure 2 - what strength of effect (quantitatively) do the colours represent?

Reply: This has been fixed.

Figures 3-5 - Grammar needs to be corrected in a few places, e.g. 'Shown is the posterior means ...'.

Reply: This has been fixed.

Reviewer Name: Thomas N O Achia

Institution and Country Division of Epidemiology and Biostatistics,

School of Public Health,

University of Witwatersrand,

Johannesburg-South Africa

Please state any competing interests or state 'None declared': None declared

The paper presents an interesting discussion concerning the association between air pollution and cardiovascular disease. The authors use a Bayesian Hierarchical spatial modeling approach to analyse the data providing an interesting way of looking at the problem. There are a few issues that need comment.

In the abstract section under primary and secondary measures, it is not clear to me what the primary outcome(s) or predictor(s) were from the statement "Air pollution, heart failure hospital admissions and mortality data within Warwickshire were" mean.

Reply: This section of the abstract has been removed as it does not apply to this type of study.

The statement "Pm [-12.93:-20.41, -6.54] was negatively associated with the risk of heart failure morbidity but with no association with So2" in the abstract is rather confusing. Are the authors suggesting that Pm was associated with So2? Currently this is what the sentence suggests and needs revision.

Reply: This sentence has been revised and split into two sentences to improve clarity.

In the abstract: "There was a striking variation in heart failure morbidity and mortality risk across wards, the highest risk being in the regions around Nuneaton and Bedworth." Please clarify what the word "a striking" means

Reply: This has been corrected in the text.

The comment: "Particulate matter (Pm) was negatively associated with the risk of heart failure morbidity" needs unpacking. Does this mean that presence of Pm is protective. If so, why?

The authors have provided a discussion around this but I am not very convinced. Could one or more of the other predictors be acting as an effect modifier? Did the authors investigate this by fitting two-way or other interaction terms? The attempt to justify the finding is not satisfactory.

There are issues here of "ecological fallacy" in nature that need to be re-examined.

Reply: The ecological nature of the study has now been highlighted in the text and the potential limitations of such an approach discussed. Some paragraphs have been added in the Discussion section which elaborates on this. The possibility that the negative correlation found between Pm air pollution and heart failure would not exist at the individual level is made clear. The potential for ecological fallacy in such a study using aggregate data from a population is considered.

Air pollution contributed an "additional rise" in the sentence just before the Methods section, does not make for good grammar.

Reply: The sentence structure has been revised.

The first sentence under the study data section should read: "In order to carry out this project, data was collected about the geographical distribution of air pollution within Warwickshire" as the placement of the comma after the word project gives a completely different meaning.

Reply: This has been corrected in the text.

It could be helpful for the authors to clarify what each of the variables or symbols in the Binomial distribution presented on line 1-11 of page 8 are.

Reply: Please see page 8. The symbols have been clarified.

VERSION 2 – REVIEW

REVIEWER	Anoop Dinesh Shah Farr Institute of Health Informatics Research United Kingdom
REVIEW RETURNED	21-Aug-2014

GENERAL COMMENTS	In the abstract, the abbreviations NOx, Ben and IMD in the abstract are ambiguous; they should be defined or written out in full. In the background, SI units should be used (kilometres rather than miles). Statistical analysis: The authors have included more detail on the Bayesian model, which is good. I assume the priors were non-informative and not based on any other background data. Pollutant should be plural in the last sentence of the results. Grammatical error in the discussion: " It is certainly possible that the unexpected negative correlation between particulate matter air pollution and heart failure could disappear when data are analysed at the individual level – an example of an ecological fallacy. "
--

VERSION 2 – AUTHOR RESPONSE

Reviewer Name Anoop Dinesh Shah
Institution and Country Farr Institute of Health Informatics Research
UCL Institute of Health Informatics
United Kingdom

Please state any competing interests or state 'None declared': None declared

In the abstract, the abbreviations NOx, Ben and IMD in the abstract are ambiguous; they should be defined or written out in full.

Reply: We have now written in full the abbreviations in the abstract as suggested: Mono-nitrogen Oxide (NOx), Benzene (Ben), sulphur dioxide (So2) and Index of Multiple Deprivation (IMD).

In the background, SI units should be used (kilometres rather than miles).

Reply: We have converted the 70 miles to 112 Kilometres as suggested.

Statistical analysis: The authors have included more detail on the Bayesian model, which is good. I assume the priors were non-informative and not based on any other background data.

Reply: we have now clarified in the text that non-informative priors were used for fixed effects parameters and they were not based on any background data.

Pollutant should be plural in the last sentence of the results.

Reply: it has been corrected.

Grammatical error in the discussion:

" It is certainly possible that the unexpected negative correlation between particulate matter air pollution and heart failure could disappear when data are analysed at the individual level – an example of an ecological fallacy. "

The authors have improved the article and it is now suitable for publication after the few corrections I have suggested.

Reply: we have deleted the word "certainly" before "possible".